# Profit or Purpose: What Increases Medical Doctors’ Job Satisfaction?

**DOI:** 10.3390/healthcare10040641

**Published:** 2022-03-29

**Authors:** Young Kyun Chang, Won-Yong Oh, Sanghee Han

**Affiliations:** 1Sogang Business School, Sogang University, 35 Baekbeom-ro, Mapo-gu, Seoul 04107, Korea; changy@sogang.ac.kr; 2Lee Business School, University of Nevada, Las Vegas, NV 89154, USA

**Keywords:** medical doctors, profit, purpose, rational choice theory, public service motivation, meaningfulness, job satisfaction

## Abstract

This study integrates two competing views to examine whether medical doctors are satisfied with their jobs when they perceive their hospitals as being oriented toward profit (i.e., rational choice theory) or purpose (i.e., public service motivation). Using a sample of 127 doctors from 70 hospitals, this study tests these competing views. The results show that doctors who perceive their hospitals as purpose-driven are likely to experience job satisfaction, and this pattern still holds even if they also perceive their hospitals to be emphasizing profits. However, only the purpose-driven orientation results in job satisfaction via a sense of meaningfulness. Thus, this study offers comprehensive evidence that while medical doctors are likely to be satisfied with their jobs when they work at either purpose-driven or profit-driven hospitals, only purpose-driven hospitals give doctors a sense of meaningfulness. This finding suggests that both rational choice theory and public service motivation perspective are valid; however, public service motivation plays a greater role in terms of a sense of meaningfulness. Theoretical contributions and practical implications are discussed.

## 1. Introduction

While healthcare systems vary across countries and regions in terms of their structures, policies, and resource availability, it is still widely recognized that medical doctors play a pivotal role in hospitals’ operations and performance of systems [1]. Medical doctors, as one of the largest groups of healthcare professionals, provide medical care and professional treatment for those who are sick and sustain the healthcare industry. Due to medical doctors’ significant roles in society, researchers have investigated their behaviors in various ways. In particular, the majority of research has accumulated knowledge and evidence, mostly of the determinants of doctors’ success and positive stories rather than exploring the amount of distress they feel and negative stories about their professional lives [2].

An increasing number of studies have departed from the conventional bright view of medical doctors, recognizing that many doctors are in fact surprisingly unhappy with their jobs. For example, in the UK, about one third of health professionals have experienced serious levels of dissatisfaction, compared to 18% of general workers; 10–20% of doctors even have reported psychological depression [3]. Similarly, evidence from other countries also indicates that doctors’ burnout is high and their overall job satisfaction is low [4,5]. Furthermore, the recent COVID-19 pandemic crisis has made doctors more frustrated with overwork, trauma, and disillusionment [6]. If doctors are not happy, how can they deliver their best and more authentic cares to patients? How, under such conditions, can healthcare and social welfare systems function well? If such a trend exists, motivating and retaining doctors would become costly and societal burdens would continue to grow. 

The purpose of the present study is, therefore, to explore what makes medical doctors satisfied with their jobs. In fact, previous studies have explored various factors that affect workers’ and professionals’ well-being and satisfaction with their jobs [7,8]. This paper examines different motivations based on rational choice theory [9] and public service motivation [10]. On one hand, a major and traditional body of research argues that compensation, rewards, and external support, albeit depending upon the context, could make workers feel happy with their jobs [11,12], as individuals have self-interests to maximize their own benefit. Healthcare researchers propose that medical doctors are not different from other professionals in how they are motivated [13]. This rational choice approach is broadly based on *the hedonic view* [14]. On the other hand, public service motivation perspective complements the rational choice theory. As with firefighters, police officers and soldiers, medical doctors are also a kind of public professional and thus likely define their vocational identity as a person who seeks a virtuous life by “serving” others and society [15]. This approach is broadly based on *the eudaimonic view* [14]. 

This study posits that such views certainly shape doctors’ mindsets and their attitudes toward work, including their job satisfaction. What matters, however, is specifically how doctors develop their job attitudes. In particular, the present study focuses on “hospital orientation”—*profit-driven* vs. *purpose-driven*—as an important organizational cue and signal that stimulates economic rational choice vs. public service motivation for doctors to develop their job attitudes. Hospitals are thought to exist for higher purposes, namely, to preserve the systems of common health and social welfare. At the same time, hospitals undeniably have a burden of economic sustainability in order to achieve those goals. Making profit is one of the key considerations for making hospitals sustainable [16,17], and only profit-generating hospitals can afford to compensate doctors adequately, ensuring that doctors enjoy their jobs and that hospitals can thus continue to operate. In other words, hospitals have to pursue both hedonic and eudaimonic goals to survive [18]. This obviously poses a dilemma for medical workers (i.e., pursuing greatest benefits vs. seeking purpose and virtue) and is an intricate context. 

This study does not take such a black-and-white approach, but rather a holistic approach. It integrates two competing views into one larger equation to better predict doctors’ job satisfaction. This study does so by examining whether medical doctors are satisfied with their jobs when they perceive their hospital to be profit-driven (i.e., a rational choice hypothesis, H1) or purpose-driven (i.e., a public service motivation hypothesis, H2). It also investigates how doctors assess their job satisfaction when the two orientations are perceived simultaneously. In addition, this study hypothesizes whether meaningfulness plays a mediating role in translating a doctor’s perception that his or her hospital is purpose-driven into job satisfaction; however, if doctors are satisfied with their jobs because of a hospital’s profit orientation, meaningfulness would not play such a role. (i.e., a mediation hypothesis, H3). Figure 1 illustrates this study’s overall research model.

This study contributes to the current literature on theory and practice. From a theoretical standpoint, it integrates two competing views (i.e., rational choice theory and public service motivation) of how and why doctors feel satisfied with their jobs. Without such a comprehensive approach, any attempt to understand doctors’ job satisfaction should be seen as partially valid, at best, or completely invalid, at worst. From a practical standpoint, this study presents feasible implications regarding how hospitals could promote their doctors’ job satisfaction. This study advises that doctors could be satisfied through genuine feelings of meaningfulness in their jobs only when hospitals are committed to a greater purpose of serving humanity. 

## 2. Theory and Hypothesis Development

An essential question in human life is how to live well (i.e., well-being) and how to find meaningfulness. This fundamental question has often been answered from two different angles—the rational choice theory (hedonic view) and public service motivation (eudaimonic view) [16]. Rational choice theory serves as the fundamental principle for behavioral science [19]. This theory focuses on the individual’s decision making to maximize total utility, and this it is hedonic in nature (i.e., people pursue for pleasure and enjoyment). In this view, human well-being is achieved through the pursuit of physical comfort, emotional pleasure, and enjoyment of social interaction, among other things [14,20]. In contrast, the public service motivation perspective argues that some individuals are motivated by and attracted to public service work, mainly due to a sense of public interest and commitment to the interests of society [21]. As such, this view suggests that well-being is not necessarily achieved through pleasure and enjoyment; rather, it is earned by seeking to develop one’s best self, including performing to the best of one’s capability, exercising virtues such as meaningfulness or gratitude, and developing one’s potential [14,22]. In a fundamental sense, those two views provide an important theoretical background to explain how and why people become happy or unhappy with their jobs in the workplace [10,23,24]. 

### 2.1. Rational Choice Thoery: A Hedonic Approach to Job Satisfaction

Based on rational choice theory [19], existing studies have claimed that job holders are satisfied when they are given decent pay and sufficient external support. It is argued that compensation and rewards are among the critical factors that affect job satisfaction, since they are closely related to one’s utility function and needs fulfillment [25]. This argument has been repeatedly confirmed in various fields and countries. For example, using a sample of job holders in the field of educational services, Miller and Lee found that top factors in producing job satisfaction are financial resources, workload, and technological impact [26]. Similarly, Rafikul and Ahmad reported that both Malaysian and American workers claimed that offering high wages is the most effective way to enhance motivation [27]. Additionally, Khojasteh suggested that for both public- and private-sector employees, a high level of reward is positively associated with higher levels of motivation, though this pattern is more salient in the private sector [28]. 

Rational choice theory could also be applied to healthcare professionals. Generous pay, reduced workload, and external support, *albeit* depending upon the context, can make healthcare professionals feel happy with their jobs, since those things make their lives easier, more comfortable, and more likely to be enjoyable [29,30,31]. For example, if healthcare professionals are paid well, they can afford to secure a certain qualify of life by being able to purchase and possess what they need. Likewise, if healthcare professionals are given resources and external support from their hospitals, they would have less stress and better access to more comfort and enjoyment. However, these can be provided when hospitals make “profits” so that they have sufficient resources to satisfy the needs of healthcare professionals. As such, it is reasonable to argue that medical doctors are likely to be satisfied with their jobs when they perceive that their hospitals are committed to making profits, since it will result in outcomes that align with their own economic benefits. As far as profits go, previous studies have claimed that organizational outcomes (e.g., profits) are often perceived by individuals as a way to make sense of relevant organizational cues [32,33]. This leads to the first hypothesis:

**Hypothesis** **1** **(H1).**
*Doctors’ perception that they work for a profit-driven hospital will be positively associated with their job satisfaction.*


### 2.2. Public Service Motivation: A Eudaimonic Approach to Job Satisfaction

The other approach to explaining how and why doctors are satisfied with their jobs is public service motivation view. Perry and Wise coined the term “public service motivation” to explain the attraction that individuals have toward public organizations and their services [34]. Public service motivation is associated with eudaimonism (i.e., doctors try to make efforts for purpose and virtue) in nature [35]. In fact, an increasing number of organizational scholars have focused on other unknown satisfaction-inducing factors that are not related to compensation or external forces, but are closely related to human potential and virtue [14,22,36]. They tend to see that a job has to be deeply embedded in an individual’s whole life and connected to a larger collectivity. Rothausen and Henderson documented that “…key meanings of work spring from basic human needs from work, and from workers’ larger worlds, and these elements likely impact workers’ satisfaction” [24] (p. 359).

Previous studies have offered some rationale for how and why individuals are satisfied with their jobs, when their motivations are based upon prosocial motives and altruism [37,38]. For example, people are satisfied with their jobs when their jobs allow them to express some important elements of themselves, including their core values and beliefs about work [24]. Similarly, job holders feel satisfied when their jobs meet their physical needs and wants by contributing to the standard of living for themselves and their larger communities [39]. Finally, job holders feel happy when their jobs play a role in a transcendent “purpose” in worthwhile communal and societal endeavors and in contributing to a larger collectivity [40]. 

Those rationales align well with why medical doctors are satisfied with their work. As with firefighters, police officers, and soldiers, doctors are public professionals and thus they are likely to define their existence to serve the society foremost. With a public-centered mindset, doctors may want to express their communal values and beliefs about work [15]. Doctors engage in treating others to fulfill their physical needs, and as such, contribute to the standard of living of the larger society. Doctors also seek purpose by pursuing worthwhile communal and societal endeavors and contributing to collectivity in a social setting. All of these conditions are exactly why hospitals should exist. Notably, this public-based claim is consistent with the accumulated findings of a large body of literature on public service motivation [10,36,41,42]. As such, if hospitals are deeply dedicated to purpose, doctors are likely to be satisfied with their jobs. This leads to the second hypothesis:

**Hypothesis** **2** **(H2).**
*Doctors’ perception that they work for a purpose-driven hospital will be positively associated with their job satisfaction.*


### 2.3. A Mediating Role of Meaningfulness

This study further investigates the hidden mediating mechanism through which a doctor’s perception that they work for a purpose-driven hospital leads to job satisfaction. This study proposes that meaningfulness plays a mediating role. Meaningfulness is the core of positive psychology, which focuses on how people prosper as human beings [43,44]. Meaningfulness is “both a cognitive and an emotional assessment of whether one’s life has purpose and value” [45] (p. 506). People feel meaningfulness from the purpose or value of their work, depending upon their psychological properties [46,47]. In the previous literature, meaningfulness has been seen as closely related to job characteristics, work motivation, and engagement at work [46,48,49]. This study extends the boundary of theorizing about meaningfulness by examining whether a doctor’s perception of a hospital orientation is associated with their sense of meaningfulness and subsequent job satisfaction. 

Recent studies have proposed that eudaimonic behaviors may lead to a job holder’s sense of meaningfulness [46,50,51]. For example, according to Colby et al., over half of a sample of Americans maintained that benefiting others is what made their work meaningful [52]. Furthermore, the transcendence of the self to others’ needs in any social group has long been recognized as fueling a sense of meaningfulness [52]. Additionally, Aguinis and Glavas argued that individuals recognize an organization’s social contribution and find meaningfulness through work, and do so by engaging in a sense-making process as an underlying mechanism [53]. In particular, Zheng and colleagues reported that employees with a high level of public service motivation are likely to experience higher meaningfulness of their work [54].

If hospitals are deeply committed to public healthcare, doctors are likely to feel a greater purpose in their vocation, and thus feel meaningfulness at work. This eventually leads to job satisfaction. Therefore, it is hypothesized that meaningfulness mediates the relationship between a doctor’s perception of a purpose-driven hospital and his or her job satisfaction. This leads to the final hypothesis:

**Hypothesis** **3** **(H3).**
*Doctors’ perception that they work for a purpose-driven hospital will be positively associated with job satisfaction via a sense of meaningfulness.*


## 3. Method

### 3.1. Sample

The survey was initially conducted with 490 doctors who were working at large hospitals in Korea. It is important to make sure that the sample properly represents the target population. We found that our sample illustrates representativeness in terms of gender and age, as shown in Table 1. Out of the 490 questionnaires distributed, 127 responses were returned (a 26% response rate). The majority of doctors were males (78%), and most of them were in their thirties or forties (20s = 7.9%, 30s = 35.7%, 40s = 47.6%, 50 and above = 8.8%). In terms of position, more than half were doctors (doctor = 66.9%, head of department = 27.6%, vice dean of hospital = 5.5%). The average work hours per week was 48, and each doctor had, on average, 40 patients under their care. The average period of tenure was six years and three months. All questionnaires were answered voluntarily and confidentially.

### 3.2. Measurement

All variables are adapted from the original questionnaires used in previous studies. Questionnaires were translated using the back-translation method [55] by two independent bilingual translators. All variables are measured based on a 7-point Likert scale, ranging from 1 (strongly disagree) to 7 (strongly agree).

Independent variables. This study introduces two paths of a doctor’s perception of their hospital’s orientation: *profit-driven* (PROF) vs. *purpose-driven* (PURP). For a profit-driven hospital, a single item is adapted by the existing measures for a corporation’s sense of economic responsibility [56] and slightly modified to accommodate hospital contexts. The item is: *My hospital focuses on a strategy that maximizes profitability in order to be sustainable.* Purpose-driven hospitals are expected to be prosocial and public-oriented, which is embedded in their identity. Three items are adapted using the existing measure of an organization’s prosocial identity [50] and modified to accommodate hospital contexts. The items are: *My hospital deeply cares for the public and society; My hospital is warm and generous toward the public and society; My hospital is recognized as an organization that contributes to the public and society* (α = 0.86). 

Dependent variable. *Job satisfaction* (JS) was measured using the classical framework of job satisfaction from Locke [50]. Six items were measured: *I am satisfied with (1) work; (2) salary; (3) the opportunity for promotion; (4) supervisor; (5) colleagues; (6) career plan in this hospital* (α = 0.89).

Mediating variable. To measure *meaningfulness* (MEAN), three items are adapted from May et al. [46]. The items are: *It is significant for me to work for this hospital; The work that I do in this hospital is very meaningful to me; The work that I do in this hospital is very valuable to me* (α = 0.94).

Control variables. Several variables were controlled to avoid alternative explanations for the relationships under study. Given that workload has a negative impact on job satisfaction, we controlled for the workload of doctors by measuring weekly working hours (WHR) as well as the number of patients under their care (PAT). Other personal characteristics variables that may affect an employee’s job attitudes and behaviors, such as age, gender, education level, position-level and tenure, were also controlled. Gender (GEN) was coded as a dummy (male = 1, female = 2). Age (AGE) was coded into a four-point scale (1 = 20s, 2 = 30s, 3 = 40s, 4 = 50 and above), and education level (EDU) was classified into two groups (1 = undergraduate degree, 2 = graduate degree). Position level (POS) was measured with three levels based on the reporting structure of hospitals (1 = doctor, 2 = head of department, 3 = vice dean of hospital). Tenure (TEN) was measured by the total number of months the doctor has been at their hospital. 

## 4. Results

### 4.1. Descriptive Statistics and Correlations

Table 2 shows descriptive statistics and correlations for all variables in this study. A doctor’s perception that their hospital is profit-driven is not significantly correlated with their perception of whether their hospital is purpose-driven. However, both perceptions are positively related to job satisfaction. Meaningfulness is more closely correlated with the purpose-driven perception than the profit-driven perception. 

### 4.2. Measurement Model Test

This study first conducted a factor analysis to confirm the discriminant validity of the measures, and then evaluated model fit indices, such as the comparative fit index (CFI), the Tucker–Lewis index (TLI), and standardized root mean square residuals (SRMR), based on the criteria suggested [57,58]. The four-factor measurement model satisfied all criteria (CFI = 0.953, TLI = 0.935, SRMR = 0.0646). Then, a four-factor model was compared to other alternative models with different factor structures. As shown in Table 3, the result of a chi-square different test indicates that the four-factor model shows a better model fit over other models. Thus, four focal variables (profit-driven, purpose-driven, meaningfulness, and job satisfaction) show discriminant validity.

### 4.3. Regression Analysis

All hypotheses were tested with an ordinary least squares regression analysis. Hypothesis 1 is based on rational choice theory and tests whether a doctor’s perception that their hospital is a profit-driven hospital is positively associated with job satisfaction. As shown in Table 4, doctors are satisfied with their jobs when they see their hospitals as being committed to profit generation (see Model 1, *β* = 0.17, *p* < 0.05). Therefore, Hypothesis 1 is supported. 

Hypothesis 2 assumes public service motivation view and tests whether a doctor’s perception that their hospital is a purpose-driven hospital is positively associated with job satisfaction. Results show that doctors are satisfied with their jobs when they see their hospitals as purpose-driven organizations (see Model 2, *β* = 0.27, *p* < 0.01). Therefore, Hypothesis 2 is also supported.

However, the results from the full model suggest, when two competing orientations jointly predict a doctor’s job satisfaction, only a purpose-driven orientation leads to job satisfaction (see full Model 3, *β* = 0.23, *p* < 0.05). 

Hypothesis 3 tests the mediating effect of meaningfulness on the relationship between a hospital’s orientation and a doctor’s job satisfaction. Following Baron and Kenny [59], Model 1 tests whether an independent variable predicts a dependent variable (i.e., a main effect); Model 2 tests whether an independent variable predicts a mediating variable; and Model 3 tests whether a mediating variable predicts a dependent variable while the main effect weakens. As shown in Table 5, meaningfulness plays a significant mediating role (see Model 3, *β* = 0.55, *p* < 0.01). Thus, Hypothesis 3 is supported. 

### 4.4. Bootstrapping Analysis

To additionally test an indirect effect of a mediating variable, a bootstrapping analysis was performed. The bootstrapping method has a non-parametric advantage and does not violate a normality assumption. As such, it can be used in a relatively smaller sample size [60]. This study used the Macro PROCESS program with mean-centered variables to test the main and mediation effect [61]. Hypotheses are tested simultaneously with 5000 bootstrap samples, and the results are considered significant if the 95% confidence interval (CI) does not include zero [62]. The result indicates that meaningfulness significantly mediates the relationship between a doctor’s perception of a purpose-driven hospital and job satisfaction (indirect effect = 0.177, *p* < 0.05; ULCI = 0.273, LLCI = 0.078).

## 5. Discussion

### 5.1. Summary of Findings

Contrary to the accumulated research evidence and the conventional positive view of medical doctors, recent studies have begun to recognize that many doctors are surprisingly dissatisfied with their jobs [3,4,5]. This is not an issue that can be taken lightly since doctors are the backbone of each country’s healthcare and welfare systems. The present study attempts to untangle one of the hidden mechanisms in how doctors become satisfied with their jobs. 

Two different theoretical perspectives—rational choice theory and public service motivation theory—were introduced to predict a doctor’s job satisfaction. Using a unique sample of 127 individual medical doctors (varying in age, tenure, number of patients under their care, education level, and position level) from 70 different hospitals (varying in type and size), this study found that doctors are satisfied with jobs either when they see that their hospitals are capable of generating profits (i.e., hedonic-based perception based on rational choice theory) or that their hospitals are dedicated to their original purpose of serving society (i.e., eudaimonic-based perception based on public service motivation view). These findings confirm that both perspectives are valid in predicting a doctor’s job satisfaction, as previous studies have documented. 

However, this study offers an additional insight that doctors are more motivated by working for a “purpose-driven” hospital when two competing orientations are jointly conceived in their assessments of their jobs. Furthermore, this psychological process is fully mediated by the perception of meaningfulness. This result clearly informs hospitals and medical institutions that doctors are satisfied with their jobs, not necessarily because their hospitals are good at profit generation, but because they feel they are included as part of socially contributing organizations (i.e., social altruism [41]). 

### 5.2. Theoretical Contributions and Practical Implications

This study makes several theoretical contributions to the literature and highlights practical implications for organizational managers in the healthcare sector. 

First, job attitudes have been a key subject for organizational researchers and theorists, but the existing studies have diverged into two competing perspectives: rational choice theory vs. public service motivation [14]. The rational choice theory, based on hedonic view, proposes that people seek greater utility, so they can live something close to an enjoyable life. Resources, external support, and monetary/non-monetary compensation are seen as sources of utility, and thus are a critical source of satisfaction with one’s job and life [14,20]. On the other hand, the public service motivation, based on the eudaimonic view, proposes that individuals have a desire to serve the benefits of others. As a result, being public agentic, harnessing one’s best self, helping others, and connecting to a larger group in society are ultimately what one needs to thrive in one’s life, thus making these factors also a critical source of satisfaction with one’s job and life, albeit in different ways [14,22]. As such, this finding is more aligned with recent studies on public service motivation [42], particularly because an organization’s orientation (i.e., a hospital’s desire to serve the public benefit) also may affect the individual’s public service motivation. 

In reality, however, people think, feel, and behave in ways that both perspectives suggest, although to varying degrees and depending upon the particular context [63,64]. As such, this study explores how doctors behave when the two perceptions co-exist in the context of hospitals. Hospitals offer an interesting empirical context in which doctors shape their perceptions of their jobs by making sense of a mixture of messages related to rational choice (e.g., *Hospitals also need money to operate; Doctors are recognized as having a high-paid profession*, etc.) and messages related to public service motivation (e.g., *Hospitals exist to serve society; Doctors should act upon a calling, not profits,* etc.). This is because no hospital is completely free from the pressure to generate profits or to be purely purpose-driven [18]. This study found that doctors are more satisfied with their jobs when they perceive their hospitals to be purpose-driven, even when the purpose orientation is jointly conceived with the profit orientation in their minds. 

For hospital managers, it should be noted that doctors do not necessarily care about how well hospitals do (i.e., profit), but rather how much hospitals do right (i.e., purpose). Furthermore, this study also suggests some broader implications for research into the healthcare profession regarding how to manage healthcare workers’ work-related stress and burnout and promote their job satisfaction and well-being.

Second, this study introduces the role of organizational orientation as a source of an employee’s work attitudes. Indeed, a number of studies have maintained that an individual’s work attitudes are not merely a matter of their own ideas, but the outcome of profound interactions between individuals and organizations [42,65]. Among the many organizational characteristics, organizational orientation has been underexplored, even though it can function as an important norm, cue, and signal that regulates employees’ attitudes and behaviors in the workplace [66]. In several studies [67,68], the notion of organizational orientation has often been juxtaposed with corporate philosophy, strategic stance, and as something that is manifested by mission statements, credos, codes of conduct, or top executives’ official messages to stakeholders. This study confirms that hospital orientation is tightly coupled with a doctor’s subjective judgment of their job. 

This finding is particularly informative for organizational managers who tend to over-focus on immediate tasks and short-term objectives as a way to engage employees with their work. Hospital managers should be advised that doctors are concerned more with who they are, what they are meant to do, and what purpose they serve in society. Therefore, managers should be careful not to create and send mixed signals to doctors in terms of what the hospital is striving for. Furthermore, managers need to reinforce the hospital’s purpose-driven culture. For example, organizations could launch a campaign of recognition and support for doctors who serve patients purposefully and mindfully, somewhat independent from their level of work or clinical performance. 

Finally, this study reveals a concealed mechanism of meaningfulness, by which doctors have positive feelings about their job in relation to the hospital’s orientation. Meaningfulness is, in a nutshell, a part of public service motivation concept, which prescribes how people thrive in their lives by serving the benefits of others [43,44]. Although the merit of meaningfulness has been widely recognized across vocations, it is still far from conclusive whether meaningfulness plays a key role in job satisfaction for those in high-paid and high-demanding jobs such as medical doctors. This study found some evidence that meaningfulness still plays a critical role in promoting job satisfaction among this kind of group. 

Hospital managers also need to be aware that doctors find their work meaningful when hospitals are purpose-driven, and this tendency still holds when workloads (e.g., work hours or number of patients under one’s care) are controlled for. This implies that meaningfulness is central to their career identity and may not be easily compromised by other external motivating schemes or forces. This insight can inform the way hospital managers might use—or refrain from using—reinforcement tactics in their efforts to encourage doctors’ feeling of meaningfulness.

### 5.3. Limitations and Future Studies

This study is not without limitations. First, this study relies on self-reporting with a cross-sectional design measuring all variables at the same time, whereas the questionnaire items could be introduced in the reverse order and randomly shuffled. This approach is still subject to common-methods variance bias (CMV) [69]. However, it should be noted that medical doctors are not easily recruited for a multi-wave survey that requires an extensive commitment by study participants. Nonetheless, future studies are encouraged to reduce CMV through objective or multisource assessments of outcomes. Furthermore, experimental studies in a laboratory setting could help in establishing causality more clearly. 

Second, in terms of measuring a perception of profit-driven hospital, a single item was adapted from the original scales in order to secure the content validity. From a psychometric test theory standpoint, this is our limitation, even though results of measurement model still showed the goodness of fit. 

Third, all participants in the current study were medical doctors, which threatens the external validity of the findings. Future studies are encouraged to examine other healthcare professionals such as nurses, clinical staff, and others, when expanding this study through the use of different samples in the healthcare industry. Having additional evidence from different healthcare professionals would enhance the external validity of the findings. 

Fourth, this study assumes that hospitals wish to make profits to keep their systems going. However, this assumption may be applicable to certain countries such as the US and Korea, but not necessarily to other countries or regions such as Canada and Europe. In fact, hospitals in some countries are not under pressure to make profits due to their unique national healthcare systems [70]. As such, it is worthwhile for future studies to investigate whether institutional characteristics (e.g., having such a national healthcare system) can make a difference in doctors’ perceptions of a hospital’s profit orientation and their job satisfaction. 

Finally, although this study found a mediating effect of meaningfulness on the relationship between doctors’ perception of a hospital’s orientation and their job satisfaction, there may be other unknown underlying variables. Researchers in the field of positive psychology may continue to explore other factors that may act as mediators, such as gratitude, pride, and calling, to name a few, to promote a doctor’s job satisfaction.

## 6. Conclusions

This study examines whether medical doctors are satisfied with their jobs when they perceive their hospitals as being oriented toward profit (i.e., rational choice theory) or purpose (i.e., public service motivation). The results, based on a sample of 127 doctors from 70 hospitals, show that doctors would be satisfied with their jobs when their hospitals are oriented towards generating profits or towards seeking a greater mission and purpose. However, only the purpose-driven orientation results in job satisfaction via a sense of meaningfulness. As such, this study concludes that both the rational choice theory and public service motivation perspective are valid; however, public service motivation plays a greater role in terms of a sense of meaningfulness. Medical doctors are satisfied with their jobs, not necessarily because their hospitals are doing well, but because they are doing right.

## Figures and Tables

**Figure 1 healthcare-10-00641-f001:**
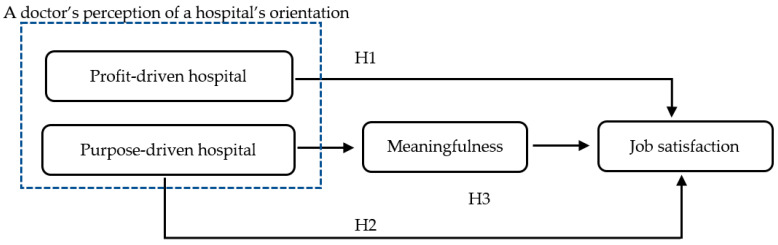
A research model of the present study.

**Table 1 healthcare-10-00641-t001:** Statistics for sample representativeness.

		Population Statistics in Korea	Sample of the Present Study
Gender	Male	76.8%	78%
	Female	23.2%	22%
Age	20s	6.0%	7.9%
	30s	32.7%	35.7%
	40s	49.4%	47.6%
	50 and above	11.9%	8.8%
Total (number of doctors)		*n* = 115,127	*n* = 127

Source: RHIP annual report from Korea Medical Association (http://www.rihp.re.kr/ accessed on 14 February 2021).

**Table 2 healthcare-10-00641-t002:** Descriptive statistics and correlations.

	M	SD	1	2	3	4	5	6	7	8	9	10
1. AGE	2.58	0.78	-									
2. GEN	1.22	0.44	−0.18 *	-								
3. EDU	1.61	0.51	0.38 **	−0.08	-							
4. POS	1.398	0.59	0.53 **	−0.18 *	0.18 *	-						
5. WHR	48.0	17.4	−0.23 **	0.08	−0.20 *	−0.11	-					
6. PAT	40.06	36.4	0.01	−0.06	0.14	−0.02	−0.10	-				
7. TEN	74.66	61.8	0.56 **	−0.07	0.38 **	0.29 **	−0.04	−0.04				
8. PROF	4.80	1.12	0.20 *	0.03	0.25 **	0.07	−0.09	−0.08	0.24 **	-		
9. PURP	4.75	0.99	0.24 **	−0.04	0.23 *	0.13	−0.27 **	0.02	0.21 *	0.14	-	
10. MEAN	5.12	1.03	0.38 **	−0.10	0.18 *	0.27 **	−0.27 **	−0.06	0.35 **	0.17 *	0.53 **	-
11. JS	4.60	0.98	0.29 **	−0.06	0.13	0.17	−0.24 **	0.08	0.32 **	0.26 **	0.39 **	0.65 **

Note: N = 127, * *p* < 0.05. ** *p* < 0.01 (two-tailed).

**Table 3 healthcare-10-00641-t003:** Results of measurement model test.

Model	χ^2^	*df*	Δχ^2^	χ^2^/*df*	CFI	TLI	SRMR
One-factor (all items combined)	431.172	54	334.887 **	7.98	0.631	0.549	0.1430
Two-factor (PROF + PURP, MEAN + JS)	232.076	52	135.801 **	4.46	0.824	0.776	0.1170
Three-factor (PROF + PURP, MEAN, JS)	132.558	50	36.283 **	2.65	0.919	0.893	0.0679
Four-factor	96.275	48	-	2.01	0.953	0.935	0.0646
Decision criteria			*** p* < 0.001		>0.90	>0.90	<0.08

**Table 4 healthcare-10-00641-t004:** Results of regression analysis on job satisfaction.

Variable				DV: Job Satisfaction					
	Model 1			Model 2			Model 3 (Full)		
	β	*SE*	T	β	*SE*	T	β	*SE*	t
*Control variables*									
Age	0.10	0.15	0.08	−0.03	0.06	−0.03	−0.02	0.15	−0.13
Gender	−0.05	0.19	−0.56	−0.09	0.09	−1.01	−0.09	0.21	−1.02
Education	−0.10	0.18	−1.02	−0.08	0.08	−0.84	−0.10	0.19	−0.99
Position	0.10	0.16	0.97	0.08	0.10	0.73	0.08	0.16	0.76
Work hours	−0.23 *	0.01	−2.48	−0.17 ^†^	0.00	−1.75	−0.17 ^†^	0.01	−1.83
Num. of patients	0.09	0.01	1.03	0.07	0.09	0.81	0.08	0.01	0.90
Tenure	0.25 *	0.01	2.29	0.25 *	0.03	2.30	0.24 *	0.01	2.22
*Testing variables*									
IV: Profit-driven	0.17 *	0.07	1.99				0.10	0.08	1.07
IV: Purpose-driven				0.27 **	0.06	2.89	0.23 *	0.10	2.41
F			3.31 **			3.88 **			3.58 **
R^2^			0.20			0.23			0.24
Adjusted R^2^			0.14			0.17			0.17

Note: N = 127, ^†^ *p* < 0.10, * *p* < 0.05. ** *p* < 0.01., IV = independent variable; DV = dependent variable.

**Table 5 healthcare-10-00641-t005:** Results of mediating effect of meaningfulness.

Variable	Model 1 (DV: JS)			Model 2 (DV: MEAN)			Model 3 (DV: JS)		
	β	*SE*	T	β	*SE*	t	β	*SE*	t
*Control variables*									
Age	−0.02	0.15	−0.13	0.11	0.15	1.04	−0.08	0.14	−0.76
Gender	−0.09	0.21	−1.02	−0.06	0.21	−0.73	−0.06	0.19	−0.78
Education	−0.10	0.19	−0.99	−0.11	0.18	−1.27	−0.04	0.17	−0.50
Position	0.08	0.16	0.76	0.12	0.16	1.25	0.02	0.15	0.17
Work hours	−0.17 ^†^	0.01	−1.83	−0.14	0.05	−1.67	−0.10	0.01	−1.21
Num. of patients	0.08	0.01	0.90	−0.01	0.01	0.04	0.08	0.01	1.02
Tenure	0.24 *	0.01	2.22	0.16	0.01	1.70	0.16 ^†^	0.01	1.68
*Testing variables*									
IV: Profit-driven	0.10	0.08	1.07	0.15	0.08	1.69	0.03	0.07	0.31
IV: Purpose-driven	0.23 *	0.10	2.41	0.36 **	0.09	4.12	0.04	0.09	0.40
MV: Meaningfulness							0.55 **	0.09	5.66
F			3.58 **			7.30 **			7.40 **
R^2^			0.24			0.39			0.43
Adjusted R^2^			0.17			0.34			0.37

Note: N = 127, ^†^ *p* < 0.10, * *p* < 0.05. ** *p* < 0.01., IV = independent variable; DV = dependent variable; MV = mediating variable.

## Data Availability

Data is available from the corresponding author by request.

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
