# Peer review of "Profit or Purpose: What Increases Medical Doctors’ Job Satisfaction?"

_healthcare, 2022, doi:10.3390/healthcare10040641_

Round 1

Reviewer 1 Report

This paper aims at identifying what are the factors that have an impact on doctors' job satisfaction when they perceive their hospitals are oriented toward profit (rational choice theory) or purpose (public service motivation). In a previous version, they referred to “hedonic view and a eudaimonic view”.

The research question is still clear and well defined. Moreover, the paper is well written and has a good literature review concerning the topic.

The Authors have also responded in a proper way to my comments.

I, therefore, believe the manuscript has been significantly improved and now warrants publication in healthcare.

Author Response

I, therefore, believe the manuscript has been significantly improved and now warrants publication in healthcare.

Response: Thank you for your positive feedback for our revised manuscript. We incorporate the second reviewer's comment. 

Reviewer 2 Report

Thank you for sending me this article for review. The article, entitled "Profit or Purpose: What Increases Medical Doctors' Job Satisfaction?", is "to explore what makes medical doctors satisfied with their jobs", as the authors point out. It is a very interesting article that highlights an aspect that influences the work environment and the coexistence of that environment in the last aspect. The following are some considerations in case they might be of interest:

  • The title is appropriate. It is brief and reflects the subject matter.
  • The abstract has an adequate structure.
  • The theoretical basis is adequate and broad. However, here it is necessary to recommend a greater use of updated references (from the last five years).
  • The methodology is also adequate.
  • The results are very well structured.
  • The discussion analyzes the hypotheses adequately.
  • It is recommended that the conclusions include more concrete information, relevant quantitative data that are simple to interpret and summarize the research.
  • The references are adequate in terms of content although it is recommended that more references from the last five years be included. 

Thank you for your work. 

Author Response

However, here it is necessary to recommend a greater use of updated references (from the last five years). The references are adequate in terms of content although it is recommended that more references from the last five years be included. 

Response: Following your recommendation, we added more recent references from the last five years, including a few papers published in Healthcare. 

It is recommended that the conclusions include more concrete information, relevant quantitative data that are simple to interpret and summarize the research.

Response: Following your recommendation, we completely rewrote the conclusion section, as you see below. 

"This study examines whether medical doctors are satisfied with their jobs when they perceive their hospitals as being oriented toward profit (i.e., rational choice theory) or purpose (i.e., public service motivation). The results, based on a sample of 127 doctors from 70 hospitals, show that doctors would be satisfied with their jobs when their hospitals are oriented to generating profits or preserving a greater mission and purpose. However, only the purpose-driven orientation results in job satisfaction via a sense of meaningfulness. As such, this study concludes that both rational choice theory and public service motivation perspective are valid; however, public service motivation plays a greater role in terms of a sense of meaningfulness. Medical doctors are satisfied with their jobs, not necessarily because their hospitals are doing well, but because they are doing right."